# Antithrombotic Strategy for Patients with Acute Coronary Syndrome: A Perspective from East Asia

**DOI:** 10.3390/jcm9061963

**Published:** 2020-06-23

**Authors:** Yohei Numasawa, Mitsuaki Sawano, Ryoma Fukuoka, Kentaro Ejiri, Toshiki Kuno, Satoshi Shoji, Shun Kohsaka

**Affiliations:** 1Department of Cardiology, Japanese Red Cross Ashikaga Hospital, Ashikaga 326-0843, Japan; 2Department of Cardiology, Keio University School of Medicine, Tokyo 160-8582, Japan; mitsuakisawano@gmail.com (M.S.); ryoma_no18@yahoo.co.jp (R.F.); sshoji0116@gmail.com (S.S.); sk@keio.jp (S.K.); 3Department of Cardiovascular Medicine, Okayama University Graduate School of Medicine, Dentistry and Pharmaceutical Sciences, Okayama 700-8558, Japan; eziken82@gmail.com; 4Department of Medicine, Icahn School of Medicine at Mount Sinai, Mount Sinai Beth Israel, New York, NY 10003, USA; kuno-toshiki@hotmail.co.jp

**Keywords:** dual antiplatelet therapy, P2Y12 inhibition, acute coronary syndrome, percutaneous coronary intervention, tailored therapy

## Abstract

Dual antiplatelet therapy (DAPT) after percutaneous coronary intervention has become the standard of care, particularly in patients with acute coronary syndrome (ACS). Current clinical guidelines recommend novel P2Y12 inhibitors (e.g., prasugrel or ticagrelor) in addition to aspirin based on the results of representative randomized controlled trials conducted predominantly in Western countries. These agents were superior to clopidogrel in reducing the composite ischemic events, with a trade-off of the increased bleeding events. However, multiple differences exist between East Asian and Western patients, especially with respect to their physique, thrombogenicity, hemorrhagic diathesis, and on-treatment platelet reactivity. Recent studies from East Asian countries (e.g., Japan or South Korea) have consistently demonstrated that use of novel P2Y12 inhibitors is associated with a higher risk of bleeding events than use of clopidogrel, despite borderline statistical difference in the incidence of composite ischemic events. Additionally, multiple studies have shown that the optimal duration of DAPT may be shorter in East Asian than Western patients. This review summarizes clinical studies of antithrombotic strategies in East Asian patients with ACS. Understanding these differences in antithrombotic strategies including DAPT and their impacts on clinical outcomes will aid in selection of the optimal tailored antithrombotic therapy for patients with ACS.

## 1. Introduction

Dual antiplatelet therapy (DAPT), namely the combination of aspirin and a P2Y12 inhibitor, has become a standard antithrombotic regimen after stent implantation in patients who undergo percutaneous coronary intervention (PCI). DAPT reduces the risk of ischemic events such as myocardial infarction and stent thrombosis [1], but it does so at the cost of an increased risk of bleeding. Hence, the current clinical practice guidelines recommend that antithrombotic therapy should be administered while seeking an optimized balance between the risks of thrombotic events and bleeding events, particularly in patients presenting with acute coronary syndrome (ACS) [2,3,4,5,6]. Recommendations regarding the selection of antithrombotic therapy in the American and European guidelines are based on clinical data from large-scale randomized controlled trials such as TRITON-TIMI 38 and PLATO [7,8,9,10]. More recently, precise bleeding risk assessments with a clear-cut definition of high bleeding risk (HBR), such as the Academic Research Consortium (ARC)-HBR criteria, have been implemented [11].

However, there are significant differences between East Asian and Western patients in terms of physique, thrombogenicity, hemorrhagic diathesis, and on-treatment platelet reactivity, particularly by clopidogrel; this is partly due to the high prevalence of the CYP2C19 loss-of-function alleles and is widely known as the “East Asian paradox” [12,13,14,15]. Furthermore, Asian subgroups have been relatively small in previous major randomized controlled trials [9,10,16]. Therefore, whether the evidence and guidelines from Western countries can be generalized to East Asian patients remains largely unclear. Given the different risk profiles between East Asian and Western populations, it is of utmost importance to reassess the Asian version of HBR criteria, the most appropriate antithrombotic regimens (especially in terms of the selection of P2Y12 inhibitors), and the optimal duration of DAPT for East Asian patients with ACS. This in-depth review mainly summarizes data from East Asian clinical studies regarding these issues and compares the data with studies conducted in Western countries. Evidence from these studies is essential for understanding the safety and efficacy of various antithrombotic strategies among East Asian patients with ACS.

## 2. Different Risk Profiles between East Asian and Western Population (“East Asian Paradox”)

As acknowledged by the World Heart Federation and Academic Research Consortium (ARC) for HBR in recent years, differences in certain characteristics exist between East Asian and Western patients with ACS, such as age, body mass index (BMI), and baseline renal function [11,12]. Nonetheless, few studies to date have been capable of detecting multiethnic disparities. East Asians are underrepresented in large-scale clinical trials and real-world databases, even in countries with diverse ethnicities such as the United States, Canada, and Australia [9,10,16]. This is a major concern in constructing an effective primary and secondary prevention strategy for East Asian patients with ACS because the ischemic and bleeding risk thresholds are thought to differ among ethnic groups [12,13].

Ethnic differences in the bleeding risk associated with warfarin use have been recognized. In one study, warfarin use was associated with an increased risk of intracranial hemorrhage among Asian patients with atrial fibrillation (AF) (hazard ratio [HR]: 4.06; 95% confidence interval [CI]: 2.47–6.65) when compared with Whites in the Kaiser Permanente Southern California, United States cohort from 1995 to 2000 [17]. More recently, in 2007, Wang et al. [18] reported that, despite similar treatment, Asian patients had a significantly higher bleeding risk than non-Asians after adjustments for BMI and renal function in the non-ST-elevation (NSTE)-ACS population derived from the CRUSADE Quality Improvement Initiative (February 2003 to December 2005) in the United States. These observations are of importance because data from both Asian and non-Asian populations were compared within the same health care system in the same country. The higher bleeding rates in Asian than American patients might be explained by the potential overdosing of antithrombotic agents because of a lower BMI and higher prevalence of chronic kidney disease (CKD) or by different responses to these medications [18]. Furthermore, in a prospective observational study of 15,603 patients from the CHARISMA trial, Mak et al. [19] demonstrated that Asians enrolled from Hong Kong, Malaysia, Singapore, and Taiwan were more susceptible to bleeding events than were Western patients with no difference in ischemic events.

After approval of clopidogrel for use in patients undergoing PCI and patients with ACS, greater attention was given to the use of point-of-care assays (e.g., VerifyNow System; Accumetrics, San Diego, CA, USA) measuring P2Y12 reaction units after a bolus dose of clopidogrel because several studies demonstrated an association between high on-treatment platelet reactivity among poor metabolizers and a higher incidence of ischemic events after PCI, including stent thrombosis [20,21]. The frequency of poor CYP2C19 metabolizers reportedly ranges from about 2% to 5% among Caucasians and Africans and is about 15% among Asians [14]. Pharmacogenomic studies have revealed that the CYP2C19 loss-of-function (*2) allele is a major driver of post-clopidogrel high on-treatment platelet reactivity [22]. Despite higher rates of high on-treatment platelet reactivity, lower rates of ischemic events and higher rates of bleeding events following clopidogrel use were observed among East Asian than Western patients; this is now commonly known as the “East Asian paradox” [12,13,15]. In a 2011 meta-analysis of 32 studies, individuals with one or more copy of CYP2C19 alleles associated with lower enzyme activity had lower levels of active clopidogrel metabolites, less platelet inhibition, a lower risk of bleeding, and a higher risk of cardiovascular disease events [23]. However, when analyses were limited to studies with 200 or more events, the point estimates were attenuated and showed no association between the genotype and clinically relevant outcomes. This finding is consistent with the effects of small-study bias. More recently, the POPular Genetics trial [24] showed that a CYP2C19 genotype-guided strategy for the selection of oral P2Y12 inhibitors was non-inferior to standard treatment with ticagrelor or prasugrel at 12 months with regard to thrombotic events and a lower rate of bleeding events among 2488 patients with ST-elevation acute myocardial infarction undergoing primary PCI recruited in European countries. Whether these results can be reproduced in East Asian countries remains unclear, but there is a reemerging expectation that similar tactics may also benefit East Asian patients.

## 3. Importance of Bleeding Risk Assessment in East Asian Patients with ACS

As noted above, DAPT provides more intense platelet inhibition than single antiplatelet therapy, resulting in incremental reductions in the risk of thrombotic events after PCI or ACS. However, DAPT has been associated with an increased risk of major bleeding [2,3,4,7,8]. This has also become more evident in the era of the novel potent P2Y12 inhibitors such as ticagrelor and prasugrel [9,10]. The ARC-HBR criteria were proposed because of increasing concern regarding bleeding events after PCI and the need for a standardized definition of HBR [11]. Consequently, HBR was defined as a ≥4% risk of Bleeding Academic Research Consortium (BARC) Grade 3 or 5 bleeding [25] or a ≥1% risk of intracranial hemorrhage at one year. Twenty clinical variables were identified as major or minor criteria by consensus, supported by published evidence, and patients are considered to have HBR if at least one major or two minor criteria are met. A previous study from a Japanese all-comers registry (CREDO-Kyoto Registry Cohort-2) [26] demonstrated that the prevalence of patients with HBR was 43% and the cumulative incidence of major bleeding in patients in the HBR group was 10.4% at one year after PCI; this is much higher than the prevalence of 4% defined at the proposal of the ARC-HBR criteria. These results suggest that application of the ARC-HBR criteria can identify patients with HBR in Japanese clinical practice despite the different risk profiles for bleeding events between East Asian and Western populations [12,13,15].

In early 2020, the Japanese version of the HBR criteria was published by the Japanese Circulation Society [27]. Within the Japanese criteria, low body weight (<55 kg for men, <50 kg for women), frailty, dialysis, heart failure, and peripheral vascular disease were included as major criteria in addition to the original ARC-HBR criteria (Table 1). Furthermore, the updated guideline stated that special attention should be given to patients aged ≥80 years based on the study results from the Japanese nationwide PCI registry [28]. This is in line with our previous report that patients undergoing PCI in Japan were of advanced age and had a lower BMI than those in the United States [29]. Despite these modifications in the HBR criteria, prasugrel has been predominantly used in Japan (approximately 70–80% of patients with ACS undergoing PCI) [30,31].

Introduction of risk prediction tools is warranted to urge improvements in practice patterns. Within the last decade, the DAPT score [2], PARIS score [3], and PRECISE-DAPT score [4] were developed in Western studies as prediction tools to identify individual patients’ risks and benefits of DAPT. Consequently, a few thrombotic and bleeding risk scores were developed in East Asian studies, such as the Asian DAPT score [5] and the CREDO-Kyoto risk score [6]. The Asian DAPT score was developed based on 13,172 patients registered in the Korean nationwide prospective registry and validated in 7529 patients registered in two large-scale randomized controlled trials from South Korea and Japan [5]. The score was reportedly useful for predicting both thrombotic and bleeding events with modest discrimination capacity. Variables that predicted the risk of bleeding events in the Asian DAPT score were older age, CKD, and anemia, which were also included in the Japanese version of the HBR criteria [27], suggesting that physicians should make efforts to minimize the bleeding risk in these patients. Further large-scale studies among all East Asian countries are needed to improve the accuracy of both thrombotic and bleeding risk prediction and explore the optimal antithrombotic therapy for the modern era of this unique population of East Asian patients with ACS undergoing PCI.

In summary, considering the differences in the risk profiles between East Asian and Western patients undergoing PCI, the Japanese version of the HBR criteria has been published. Although this may be useful in East Asian countries for assessment of the bleeding risk in patients undergoing PCI, these HBR criteria need to be modified or updated in accordance with the ongoing changes in the eras and environments of each country.

## 4. Optimal Antithrombotic Regimen in East Asian Patients with ACS

As in Western countries, clopidogrel, in addition to aspirin, has been traditionally used after stent implantation in East Asian patients with ACS [32,33,34]. More recently, prasugrel and ticagrelor have also been introduced as potent P2Y12 inhibitors [30,31,35]. Clinical studies regarding the selection of P2Y12 inhibitors among patients with ACS conducted in East Asian countries are summarized in Table 2.

Large-scale randomized controlled trials (TRITON-TIMI 38 and PLATO) have demonstrated that compared with the use of clopidogrel, the use of potent P2Y12 inhibitors such as prasugrel (loading/maintenance doses of 60/10 mg daily) or ticagrelor (loading dose of 180 mg followed by 90 mg twice daily) is associated with a lower risk of ischemic events but a higher risk of bleeding events among patients with ACS [9,10]. Prasugrel is a new-generation thienopyridine antiplatelet agent providing prompter and more potent platelet inhibition than clopidogrel because of its simpler metabolic pathway to become an active agent [9,36,44,45,46]. Prasugrel is also considered to be less susceptible to CYP2C19 polymorphisms, which is more common in East Asian than Western patients [12,13,14]. Multiple clinical studies from Japan and South Korea have compared prasugrel and clopidogrel among East Asian patients with ACS. The PRASFIT-ACS study [36] was a randomized controlled trial conducted in Japan to investigate the safety and efficacy of low-dose prasugrel at loading/maintenance doses of 20/3.75 mg among 1363 Japanese patients with ACS. The study demonstrated that use of low-dose prasugrel was associated with a lower incidence of composite ischemic events compared with a standard clopidogrel dosing regimen at 24 weeks (9.4% vs. 11.8%; HR: 0.77; 95% CI: 0.56–1.07), although the difference was not significant mainly because of the limited number of patients. Importantly, the incidence of major bleeding was comparable between the two regimens (1.9% vs. 2.2%; HR: 0.82; 95% CI: 0.39–1.73). Based on the results of the study, this adjusted-dose regimen of prasugrel was approved in Japan in 2014. However, the PRASFIT-ACS study was not an all-comers study, and the generalizability of the study results to all Japanese patients with ACS remains debatable. Thus, outcome data of low-dose prasugrel compared with clopidogrel should have been collected in the post-approval phase from large-scale real-world multicenter PCI registries. Two retrospective studies were recently performed to address this issue. In a Japanese contemporary multicenter PCI registry (JCD-KiCS registry), Shoji et al. [30] demonstrated that administration of low-dose prasugrel was significantly associated with a higher risk of short-term bleeding events than was clopidogrel (odds ratio [OR]: 2.91; 95% CI: 1.63–5.18; *p* < 0.001) after propensity score matching (901 pairs) in Japanese patients with ACS who underwent PCI. However, there was no significant difference in the incidence of composite ischemic events between the two groups (OR: 1.42; 95% CI: 0.90–2.23). The higher bleeding tendency with the use of low-dose prasugrel than clopidogrel was observed especially in older patients (>75 years of age), female patients, those with low body weight (<60 kg), prior cerebrovascular disease, CKD, and ST-elevation acute myocardial infarction. Similarly, in a Japanese nationwide PCI registry (J-PCI registry) that included 62,737 Japanese patients with ACS who underwent PCI, Akita et al. [31] demonstrated that low-dose prasugrel was significantly associated with a higher risk of in-hospital bleeding complications requiring blood transfusion than was clopidogrel (OR: 1.65; 95% CI: 1.10–2.51; *p* = 0.016). However, there was no significant difference in the occurrence of stent thrombosis or in-hospital mortality between the two different antiplatelet regimens. Notably, low-dose prasugrel was used in approximately 70% of the patients in this study, and the difference in the bleeding rate between the two groups was mainly derived from access-site bleeding (OR: 2.21; 95% CI: 1.31–3.89; *p* = 0.004). Therefore, Japanese real-world data consistently show that the use of prasugrel (even at the adjusted low dose) was associated with a higher bleeding risk than was clopidogrel.

Ticagrelor is a new adenosine diphosphate receptor antagonist providing prompter and more potent platelet inhibition than clopidogrel, and it is less susceptible to CYP2C19 polymorphisms [10,37,47,48]. Because ticagrelor itself is an active antiplatelet agent, it does not require an activation process, unlike clopidogrel and prasugrel. Several clinical studies from East Asian countries (Japan, South Korea, China, and Taiwan) have compared ticagrelor with clopidogrel. The PHILO study [37], a randomized controlled trial performed in Japan, South Korea, and Taiwan to investigate the safety and efficacy of ticagrelor among 801 patients with ACS, demonstrated that the use of ticagrelor was associated with a numerically higher but statistically insignificant incidence of both composite ischemic events (9.0% vs. 6.3%; HR: 1.47; 95% CI: 0.88–2.44) and bleeding events (10.3% vs. 6.8%; HR: 1.54; 95% CI: 0.94–2.53) compared with clopidogrel at 12 months. In Japan, based on the results of the PHILO study, ticagrelor is used in patients with ACS only when prasugrel or clopidogrel is unavailable or contraindicated [27]. In a Korean multicenter PCI registry (KAMIR-NIH), Park et al. [38] reported that the use of ticagrelor was significantly associated with a higher risk of in-hospital major bleeding events than was clopidogrel (2.6% vs. 1.2%; HR: 1.971 95% CI: 1.086–3.577; *p* = 0.008) after propensity score matching (1377 pairs) in Korean patients who presented with acute myocardial infarction (AMI) and underwent successful PCI. However, there was no significant difference in composite ischemic events at six months (4.2% vs. 4.9%; HR: 0.784; 95% CI: 0.491–1.253). In particular, an increased risk of bleeding with the use of ticagrelor was observed in older patients (>75 years of age), patients with low body weight (<60 kg), and those who underwent a transfemoral intervention. The TICA-KOREA study [39], a randomized controlled trial comparing the rates of ischemic and bleeding events between ticagrelor and clopidogrel among 800 Korean patients with ACS, demonstrated that the use of ticagrelor was significantly associated with a higher incidence of bleeding events at 12 months (11.7% vs. 5.3%; HR: 2.26; 95% CI: 1.34–3.79; *p* = 0.002). In addition, there was a borderline significant difference in the incidence of composite ischemic events between the two regimens (9.2% vs. 5.8%; HR: 1.62; 95% CI: 0.96–2.74; *p* = 0.07).

In contrast to the studies from Japan and South Korea, three Chinese and Taiwanese studies demonstrated superior results of ticagrelor over clopidogrel. The interim analysis of the COSTIC trial [40], which is an ongoing, prospective, observational, Chinese single-center study, demonstrated that the use of ticagrelor was associated with a lower incidence of composite ischemic events than was clopidogrel at one month (0.6% vs. 1.4%; HR: 0.44; 95% CI: 0.22–0.89; *p* = 0.019), especially in patients who presented with AMI. However, the incidence was similar at 6 and 12 months after propensity score matching (1833 pairs), with a higher incidence of BARC Type 2 bleeding [25] in the ticagrelor group. The ESTATE trial [41], a Taiwanese retrospective pilot study that investigated the safety and efficacy of ticagrelor compared with clopidogrel among 928 consecutive patients with ACS, demonstrated that the use of ticagrelor was associated with a lower incidence of composite ischemic events (HR: 0.56; 95% CI: 0.30–1.04; *p* = 0.07) and stroke (HR: 0.15; 95% CI: 0.02–1.24; *p* = 0.08) with borderline statistical significance after propensity score matching (224 pairs). The incidence of bleeding events was comparable between the two regimens, although more patients developed dyspnea and consequent drug discontinuation in the ticagrelor group than clopidogrel group (21.0% vs. 11.6%; *p* = 0.007). Finally, a Taiwanese nationwide cohort study of 27,339 patients who presented with AMI demonstrated that the use of ticagrelor was associated with a lower incidence of composite ischemic events than was clopidogrel at 18 months (10.6% vs. 16.2%; HR: 0.779; 95% CI: 0.684–0.887), without an increased risk of bleeding events (3.2% vs. 4.1%; HR: 0.731; 95% CI: 0.522–1.026) after 1:8 propensity score matching [42].

More recently, some clinical studies from South Korea have comprehensively compared clopidogrel and potent P2Y12 inhibitors such as prasugrel and ticagrelor. In a Korean nationwide registry-based study of patients with AMI (KAMIR-NIH), Kang et al. [43] reported that the use of potent P2Y12 inhibitors such as standard-dose prasugrel (HR: 2.14; 95% CI: 1.53–2.99; *p* < 0.001) and ticagrelor (HR: 2.26; 95% CI: 1.73–2.95; *p* < 0.001) was significantly associated with a higher bleeding risk than clopidogrel during a one-year follow-up after the index PCI for the ACS events, although there was no significant difference in the one-year composite ischemic outcomes among the three different antiplatelet regimens. Another Korean nationwide cohort study of 70,715 patients with ACS demonstrated that, compared with clopidogrel, the use of ticagrelor was significantly associated with a higher incidence of bleeding at two years (18.1% vs. 15.1% HR: 1.23; 95% CI: 1.14–1.33; *p* < 0.001) and lower mortality rate (3.1% vs. 3.9%; HR: 0.76; 95% CI: 0.63–0.91; *p* = 0.002) as well as no significant difference in composite ischemic events (13.1% vs. 13.0%; HR: 1.00; 95% CI: 0.92–1.09) [35]. Furthermore, compared with clopidogrel, the use of prasugrel was significantly associated with a higher incidence of bleeding at two years (14.8% vs. 12.5%; HR: 1.23; 95% CI: 1.06–1.43; *p* = 0.01), whereas the incidence of composite ischemic events (10.3% vs. 11.4%; HR: 0.88; 95% CI: 0.74–1.05) and the mortality rate (1.6% vs. 1.9%; HR: 0.78; 95% CI: 0.50–1.22) were similar between the two regimens. In this study, the risks of ischemic events and bleeding events were comparable between prasugrel and ticagrelor.

In summary, most studies conducted in East Asian countries, especially Japan and South Korea, consistently showed that the use of potent P2Y12 inhibitors such as prasugrel (even at the Japanese adjusted dose) and ticagrelor in East Asian patients with ACS was associated with a higher risk of bleeding than was clopidogrel, despite the presence of only a borderline statistically significant difference or no significant difference in the rates of composite ischemic events including mortality, myocardial infarction, stent thrombosis, or stroke. These important study results should be taken into account especially in East Asian countries, where there are larger numbers of advanced-age and underweight patients with ACS than in Western countries [28,29,49,50]. However, because Chinese and Taiwanese studies have demonstrated that ticagrelor is superior to clopidogrel in reducing ischemic events without an increased risk of bleeding [40,41,42], further studies are warranted to determine the optimal antithrombotic regimen in East Asian patients with ACS. At present, based on the results of previous studies, a tailored antiplatelet regimen with appropriate dosages should be considered to balance the risks of both ischemic and bleeding events among East Asian patients with ACS [27,51].

## 5. Optimal Duration of DAPT in East Asian Patients with ACS

With respect to the duration of DAPT, previous studies have shown a low risk of ischemic events and a high risk of bleeding events in East Asian patients who underwent PCI, and a short DAPT duration should be recommended even in patients with ACS who have a high risk of ischemic events [11,52,53]. In a meta-analysis of 13 randomized controlled trials (five East Asian studies and eight non-East Asian studies), Ki et al. [52] demonstrated that a long DAPT duration was significantly associated with a lower incidence of ischemic events than a short DAPT duration in non-East Asian patients with coronary artery disease, while the effect was neutral in East Asian patients. Furthermore, a long DAPT duration was significantly associated with a higher incidence of bleeding events than a short DAPT duration in both East Asian and non-East Asian patients. Similarly, in a landmark analysis from the time of DAPT discontinuation in the short DAPT group, Kang et al. [53] compared a prolonged DAPT group with a single antiplatelet therapy group in East Asian and non-East Asian patients using data from seven randomized controlled trials. The meta-analysis showed no effect of prolonged DAPT in reducing the ischemic risk in East Asian patients despite the significantly increased risk of bleeding. Therefore, to prevent ischemic events after PCI, a long DAPT duration is promising in non-East Asian patients but not in East Asian patients. Although patients both with and without ACS were included in each randomized controlled trial, high proportions of patients with ACS (30–70%) were evaluated in these meta-analyses. Therefore, these results are considered applicable to some extent in patients with ACS. The SMART-DATE study [54], a randomized controlled trial of patients with ACS not included in these meta-analyses, also demonstrated non-inferiority of composite ischemic events and bleeding with six months of short DAPT compared with 12 months of long DAPT. Therefore, it is reasonable for East Asians, even those with ACS, to apply a short DAPT duration rather than a long DAPT duration. Additionally, with respect to the specific DAPT duration, the administration period has differed (3, 6, or 12 months) depending on the trials, but the most sufficient evidence to date supports a DAPT duration of six months [54,55]. The STOPDAPT-2 trial [56], which validated a very short DAPT of 1 month in Japan, revealed non-inferiority in the incidence of composite ischemic cardiovascular events and superiority in the incidence of bleeding in the very-short-duration DAPT group compared with the 12-month-duration DAPT group. Approximately 35% of the study population had ACS, but no interaction was observed in the subgroup analyses between patients with and without ACS. However, this study had several important limitations: not all patients had ACS, all patients were treated with cobalt-chromium drug-eluting stents, and the only post-DAPT antiplatelet agent was clopidogrel. In addition to these important clinical trials, further studies regarding the optimal duration of DAPT in East Asian patients with ACS are warranted. A tailored antiplatelet strategy in East Asian patients with ACS should be considered based on the individual risk assessment of both ischemic and bleeding events.

## 6. Optimal Antithrombotic Regimen in Patients with AF Undergoing PCI

Antithrombotic agents are necessary for thromboembolic risk protection in patients with AF undergoing PCI and/or for those presenting with ACS. Although triple antithrombotic therapy (TAT) such as DAPT plus an oral anticoagulant (OAC) can increase the bleeding risk, stroke and stent thrombosis prevention is crucial for these patients [57,58]. Recent randomized controlled trials demonstrated favorable outcomes of patients with AF and coronary artery disease treated by dual antithrombotic therapy (DAT) including P2Y12 inhibitors and direct OACs (DOACs) [59,60,61,62]. In addition, a recent meta-analysis of these representative trials demonstrated that combined antithrombotic strategies with DOACs (as DAT or TAT) were significantly associated with a lower incidence of bleeding events than those with vitamin K antagonist (VKA), without an increased risk of ischemic events [63]. These results are consistent with the outcomes of patients with ACS from the AUGUSTUS trial [64]. Moreover, according to the Japanese and East Asian subgroup analysis of the REDUAL-PCI trial, DAT may also be allowed in patients with AF undergoing PCI [65]. Based on these results, the most recent American guideline recommends the combination of an OAC and a P2Y12 inhibitor as DAT in patients with AF who undergo PCI with stenting for ACS [66], although the European Society of Cardiology states that one month of TAT can be considered based on the patient’s bleeding and ischemic risks [8]. Therefore, the decision of whether to administer DAT or TAT must be based on the patient’s ischemic versus bleeding risk. In addition, a recent network meta-analysis [67] demonstrated that apixaban-based DAT might be the best option for patients undergoing PCI who need anticoagulation for AF in terms of reducing the risk of bleeding events, which might be reasonable especially for East Asian patients given their HBR profiles [53]. Notwithstanding these benefits, there are several issues associated with the use of DAT for patients with AF undergoing PCI. Because a previous meta-analysis revealed that DAT might increase the risk of stent thrombosis [68], DAT should be selected based on the patient’s risk of stent thrombosis, which is indicated by information such as the patient’s angiographic data (e.g., bifurcation lesions or chronic total occlusion lesions) and clinical data (e.g., presence of ACS or diabetes mellitus) [69]. Although a meta-analysis of patients with ACS and AF undergoing PCI also revealed that the use of DAT was significantly associated with a decreased risk of bleeding without an increase in ischemic events compared with TAT [70], this finding should be interpreted with caution because the incidence of stent thrombosis was not analyzed. However, in general, it is noteworthy that the risk of stent thrombosis is low with both DAT or TAT, but the long-term TAT is significantly associated with an increased risk of bleeding events [59,60,61,62].

The post-hoc analysis of the AUGUSTUS trial demonstrated an increased risk of bleeding with aspirin in addition to DAT using an OAC and P2Y12 inhibitor after 30 days of randomization without a decreased risk of ischemic events [71]. However, with regard to 30-day short-term outcomes, this study also demonstrated that aspirin could decrease the risk of severe ischemic events despite an increased risk of severe bleeding. In addition, the AUGUSTUS trial allowed randomization within 14 days after PCI [61], which might have mitigated the effect of TAT in the early phase. Based on these findings, physicians might need to decide whether to administer DAT or 1 month of TAT in patients with AF undergoing PCI.

There are several concerns associated with DAT, especially for the East Asian population. Because some East Asian patients poorly metabolize clopidogrel, DAT including clopidogrel might be challenging [72]. Although there are no data comparing more potent P2Y12 inhibitors (prasugrel or ticagrelor) versus clopidogrel as DAT regimens, and although most previous trials included clopidogrel in approximately 90% of treatments [59,60,61,62], potent P2Y12 inhibitors might be better for DAT in East Asian patients with ACS because of their high prevalence of CYP2C19 loss-of-function alleles [14]. In contrast, the current guidelines recommend clopidogrel rather than a potent P2Y12 inhibitor if the patient needs TAT [8]. Recent data from a Japanese multicenter registry-based study showed that low-dose prasugrel (3.75 mg/day) as part of TAT did not increase the risk of bleeding compared with clopidogrel-based TAT [73]. Thus, low-dose prasugrel might be a reasonable option for East Asian patients with AF undergoing PCI.

A Korean retrospective cohort study demonstrated that VKA-based TAT reduced major adverse cardiac and cerebrovascular events compared with conventional DAPT (aspirin and clopidogrel) in patients with AF undergoing coronary artery stenting, without an increased risk of bleeding [74]. The authors considered that this might have been partly due to the relatively low prothrombin time–international normalized ratio (PT-INR) control (1.83 ± 0.41) in the study patients. Moreover, a nationwide Korean study showed the gradual increase in the prescription rate of TAT for patients with AF undergoing PCI [75]. Because the Japanese guideline recommends that a target PT-INR of 1.6–2.6 for patients with AF aged >70 years [76,77], a lower therapeutic level of warfarin with DAPT might be another option, especially for patients at high risk of thrombotic events including stent thrombosis.

Adherence to medication is another issue, especially for patients undergoing DOAC-based DAT. DOAC compliance during treatment with a DAT regimen is extremely crucial in the early phase after PCI because the half-life of DOACs is relatively shorter than that of antiplatelets [78]. In addition, data from Japan, which has national medical insurance, showed that frequent daily dosing of anticoagulants was an independent risk factor for non-adherence [79]. If patients have issues with adherence, DAT with a DOAC in the early phase after PCI might be dangerous, and VKA-based DAT might be a better option because of medication adherence/compliance problems in aging East Asian societies [80,81].

In summary, although recent data have shown that DAT is better than TAT in terms of reducing the bleeding risk without an increased risk of ischemic events, stent thrombosis remains a critical issue. Physicians must decide which is better: DAT or TAT, DOAC or VKA, and clopidogrel or ticagrelor/prasugrel; they must also determine when to drop aspirin (immediately after, 14 days after, or 30 days after PCI) based on the patient’s ischemic versus bleeding risk profile.

## 7. Optimal Timing of Administering P2Y12 Inhibitors in East Asian Patients with NSTE-ACS

The American and European guidelines recommend against administering prasugrel in patients with NSTE-ACS in whom the coronary anatomy is not known [82,83]. This recommendation is based on the results of the ACCOAST trial [84], in which pretreatment with prasugrel in patients with NSTE-ACS who were planned to undergo cardiac catheterization did not reduce the incidence of ischemic events up to 30 days, but increased the incidence of major bleeding compared with the control group. In Japan, both protocol of the randomized controlled trial and the clinical practice guideline have recommended administration of P2Y12 inhibitors before PCI [27,36], but there is no elaboration on the optimal timing for pretreatment with prasugrel—at the time of diagnosis of ACS or immediately after diagnostic angiography—for patients with NSTE-ACS undergoing PCI. In fact, in Japan, P2Y12 inhibitors (low-dose prasugrel or clopidogrel) are often administered at the time of ACS diagnosis before the coronary anatomy is confirmed because coronary artery bypass grafting is rarely performed for patients with ACS due to patient and institution disfavor of surgical procedures. Furthermore, a recent study of pretreatment with prasugrel for Japanese patients with NSTE-ACS demonstrated that the administration of prasugrel at the time of diagnosis of ACS was associated with significantly reduced in-stent thrombus and/or plaque protrusion immediately after PCI as compared with the administration of prasugrel after the coronary angiography prior to PCI, suggesting that administration of prasugrel at the time of diagnosis of ACS is the more favorable choice [85].

However, some studies showed that the use of prasugrel, even at a lower dose, was associated with an increased risk of bleeding compared with clopidogrel [30,31], raising questions regarding whether prasugrel administration should be restricted to patients at low bleeding risk or whether prasugrel should not be used before diagnostic angiography. Further studies regarding the timing and safety of prasugrel administration are warranted to assess the optimal timing of pretreatment with P2Y12 inhibitors for East Asian patients who present with NSTE-ACS.

## 8. Conclusions

This review summarizes the differences in patients’ characteristics, thrombotic and bleeding risks, and optimal antithrombotic strategies, especially in the selection of P2Y12 inhibitors and the optimal DAPT duration between East Asian and Western patients. Understanding these differences may contribute to the selection of optimal tailored antithrombotic therapy and improvement in the quality of medical care for both East Asian and non-East Asian patients who present with ACS.

## Figures and Tables

**Table 1 jcm-09-01963-t001:** Major and minor criteria for high bleeding risk defined by ARC and JCS.

Major	Minor
	● Age of ≥75 years
※ Body weight of <55 kg for men and <50 kg for women ※ Frailty	
● Severe or end-stage CKD (eGFR of <30 mL/min) ※ Dialysis	● Moderate CKD (eGFR of 30–59 mL/min)
● Hb of <11 g/dL	● Hb of 11.0–12.9 g/dL for men and 11.0–11.9 g/dL for women
※ Heart failure	
● Long-term use of anticoagulation	● Long-term use of NSAIDs or steroids
※ Peripheral vascular disease	
● Spontaneous (non-intracranial) bleeding requiring hospitalization or transfusion in the past 6 months or at any time if recurrent	● First spontaneous (non-intracranial) bleed requiring hospitalization or transfusion in the past 6–12 months
● Previous spontaneous intracranial hemorrhage at any time ● Previous traumatic intracranial hemorrhage within the past 12 months ● Presence of a brain arteriovenous malformation ● Moderate or severe ischemic stroke within the past 6 months	● Previous ischemic stroke not meeting the major criteria
● Thrombocytopenia (platelet count of <100 × 10^9^/L)	
● Active malignancy within the past 12 months	
● Liver cirrhosis with portal hypertension	
● Chronic bleeding diathesis	
● Nondeferrable major surgery on DAPT	
● Recent major surgery or major trauma within 30 days	

● Common criteria in both ARC and JCS; ※ Unique criteria in JCS. Patients are considered to have a high bleeding risk if at least one major or two minor criteria are met in both the ARC and JCS versions. ARC, Academic Research Consortium; CKD, chronic kidney disease; DAPT, dual antiplatelet therapy; eGFR, estimated glomerular filtration rate; Hb, hemoglobin; JCS, Japanese Circulation Society; NSAIDs, nonsteroidal anti-inflammatory drugs.

**Table 2 jcm-09-01963-t002:** Major clinical studies regarding the selection of P2Y12 inhibitors among East Asian patients with ACS.

Name of Study or Database	Country/Year	Study Design	Study Population	Sample Size	Main Findings
		**Clopidogrel vs. prasugrel**			
PRASFIT-ACS [36]	Japan/2014	RCT; clopidogrel vs. low-dose prasugrel (loading/maintenance doses of 20/3.75 mg)	Patients with ACS undergoing PCI	*n* = 1363	Low-dose prasugrel was associated with a low incidence of composite ischemic events compared with clopidogrel at 24 weeks (HR: 0.77; 95% CI: 0.56–1.07), with a similar rate of TIMI major bleeding (HR: 0.82; 95% CI: 0.39–1.73).
KiCS-PCI registry [30]	Japan/2019	Observational multicenter registry-based study using propensity score matching; clopidogrel vs. low-dose prasugrel (loading/maintenance doses of 20/3.75 mg)	Patients with ACS undergoing PCI	*n* = 2770 (901 pairs after propensity score matching)	Low-dose prasugrel was associated with a higher risk of short-term bleeding events than was clopidogrel (OR: 2.91; 95% CI: 1.63–5.18), although there was no significant difference in the incidence of composite ischemic events between the two groups (OR: 1.42; 95% CI: 0.90–2.23).
J-PCI registry [31]	Japan/2019	Observational nationwide registry-based study using propensity score matching; clopidogrel vs. low-dose prasugrel (loading/maintenance doses of 20/3.75 mg)	Patients with ACS undergoing PCI	*n* = 62,737 (12,016 pairs after propensity score matching)	Low-dose prasugrel was associated with a higher risk of in-hospital bleeding complications requiring blood transfusion than was clopidogrel (OR: 1.65; 95% CI: 1.10–2.51), although there was no significant difference in in-hospital mortality or the occurrence of stent thrombosis between the two groups.
		**Clopidogrel vs. ticagrelor**			
PHILO [37]	Japan, South Korea, and Taiwan/2015	RCT; clopidogrel vs. ticagrelor	Patients with ACS who were planned to undergo PCI	*n* = 801	Ticagrelor was associated with a numerically higher but statistically insignificant incidence of both ischemic events (HR: 1.47; 95% CI: 0.88–2.44) and bleeding events (HR: 1.54; 95% CI: 0.94–2.53) compared with clopidogrel at 12 months.
KAMIR-NIH [38]	South Korea/2016	Observational multicenter registry-based study using propensity score matching; clopidogrel vs. ticagrelor	Patients with AMI who underwent successful PCI	*n* = 8010 (1377 pairs after propensity score matching)	Ticagrelor was associated with a higher risk of in-hospital TIMI major bleeding than was clopidogrel (HR: 1.971; 95% CI: 1.086–3.577) despite the lack of a significant difference in the composite ischemic events at 6 months (HR: 0.784; 95% CI: 0.491–1.253).
TICA-KOREA [39]	South Korea/2019	RCT; clopidogrel vs. ticagrelor	Patients with ACS who were planned to undergo invasive management	*n* = 800	Ticagrelor was associated with a higher incidence of bleeding events than clopidogrel at 12 months (HR: 2.26; 95% CI: 1.34–3.79) and a numerically higher but statistically insignificant incidence of composite ischemic events (HR: 1.62; 95% CI: 0.96–2.74).
COSTIC [40]	China/2019	Prospective observational single-center study using propensity score matching; clopidogrel vs. ticagrelor	Patients with ACS undergoing PCI	*n* = 4465 (1833 pairs after propensity score matching)	Ticagrelor was associated with a lower incidence of composite ischemic events than clopidogrel at 1 month (0.6% vs. 1.4%) (HR: 0.44; 95% CI: 0.22–0.89), especially in patients who presented with AMI, but the incidence was similar at 6 and 12 months. The incidence of BARC type 2 bleeding was consistently higher in the ticagrelor group than clopidogrel group at 1, 6, and 12 months.
ESTATE [41]	Taiwan/2016	Observational multicenter pilot study using propensity score matching; clopidogrel vs. ticagrelor	Patients with ACS who received DAPT (aspirin and one P2Y12 inhibitor)	*n* = 928 (224 pairs after propensity score matching)	Ticagrelor was associated with a lower incidence of composite ischemic endpoints (HR: 0.56; 95% CI: 0.30–1.04) and stroke (HR: 0.15; 95% CI: 0.02–1.24) with borderline statistical significance compared with clopidogrel, whereas the incidence of bleeding events was comparable between the two regimens.
Taiwan National Health Insurance Research Database [42]	Taiwan/2018	Observational nationwide registry-based study using propensity score matching; clopidogrel vs. ticagrelor	Patients with AMI who received DAPT and survived more than 30 days	*n* = 27,339 (2389 in ticagrelor group and 19,112 in clopidogrel group after 1:8 propensity score matching)	Ticagrelor was associated with a lower incidence of composite ischemic outcomes than clopidogrel at 18 months (HR: 0.779; 95% CI: 0.684–0.887), without an increased risk of bleeding events (HR: 0.731; 95% CI: 0.522–1.026).
		**Clopidogrel vs. prasugrel or ticagrelor**			
KAMIR-NIH [43]	South Korea/2018	Observational multicenter registry-based study; clopidogrel vs. prasugrel or ticagrelor	Patients with ACS who received DAPT (aspirin and one P2Y12 inhibitor)	*n* = 9355 (6444 in clopidogrel group, 1811 in ticagrelor group, 1100 in prasugrel group)	Standard-dose prasugrel (HR: 2.14; 95% CI: 1.53–2.99) and ticagrelor (HR: 2.26; 95% CI: 1.73–2.95) were associated with a higher bleeding risk than was clopidogrel during a 1-year follow-up, although there was no significant difference in composite ischemic outcomes among the three antiplatelet regimens.
Databases of the National Health Insurance Service (NHIS) [35]	South Korea/2019	Observational nationwide cohort study; clopidogrel vs. prasugrel or ticagrelor using propensity score matching	Patients with ACS treated with newly initiated P2Y12 inhibitors	*n* = 70,715 (56,216 in clopidogrel group, 11,402 in ticagrelor group, 3097 in prasugrel group)	Ticagrelor was associated with a higher incidence of bleeding events at 2 years (HR: 1.23; 95% CI: 1.14–1.33) and lower mortality rate (HR: 0.76; 95% CI: 0.63–0.91) as well as no significant difference in composite ischemic events (HR: 1.00; 95% CI: 0.92–1.09) compared with clopidogrel. Prasugrel was associated with a higher incidence of bleeding events at 2 years than was clopidogrel (HR: 1.23; 95% CI: 1.06–1.43), whereas the incidence of composite ischemic events (HR: 0.88; 95% CI: 0.74–1.05) and all-cause mortality (HR: 0.78; 95% CI: 0.50–1.22) was similar.

RCT, randomized controlled trial; ACS, acute coronary syndrome; PCI, percutaneous coronary intervention; HR, hazard ratio; CI, confidence interval; OR, odds ratio; AMI, acute myocardial infarction; TIMI, Thrombolysis In Myocardial Infarction; BARC, Bleeding Academic Research Consortium.

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
