# Peer review of "Antithrombotic Strategy for Patients with Acute Coronary Syndrome: A Perspective from East Asia"

_jcm, 2020, doi:10.3390/jcm9061963_

Round 1

Reviewer 1 Report

I have to congratulate the authors for this interesting and comprehensive review of the topic. I have no specific suggestions to improve the paper and conclusions. Concerning, DAT vs TAT and the incidence of stent thrombosis, you might comment that generally the risk of stent thrombosis is low with both regimens, but rate of bleeding events is high and significanlty higher when using long TAT.

Reviewer 2 Report

This is a very well written review.  It may add credence to the review if it stated it was a systematic review that followed PRISMA guidelines.

An overall issue to address: why were meta-analysis techniques not used, because of the small number of studies?

Also, it would be more interesting to read the paper if the tables were included in the paper itself, not as supplementary materials.

Line 27: Dual antiplatelet therapy (DAPT) would assist the reader.

Line 134: Define high bleeding risk as HBR at first occurrence.

Line 137: The Academic Research Consortium (ARC)- define first.

Line 170: Antithrmbotic -- spelling.

Lines 210-211: "...significant difference in in-hospital mortality or the occurrence of stent thrombosis between the two different antiplatelet...." would read better as "significant difference in the occurrence of stent thrombosis or in-hospital mortality between the two different antiplatelet..."

Line 218: "... CYP2C19 polymorphisms [...." because there is more than one 2C19 polymorphism.

Line 286: "where there are larger numbers of advanced-age...."

Lines 380-381: This sentence is unclear: "Moreover, a nationwide Korean study showed a recent increment of TAT for these patients [74]."

Line 391: minor point: "medication adherence problems" - probably should say adherence/compliance because you don't know which is the deficiency here.

Line 398: "...especially important for East Asian patients." Why? because of HBR? or something more specific such as 2C19 polymorphisms?

Reviewer 3 Report

In this article from Numasawa et al., the authors present a review on antithrombotic strategy for patients with Acute Coronary Syndrome, particularly focusing on Asian patients.

The paper is well written. Each part of the article is almost well balanced with each other, since bleeding risk seems to acquire much more dignity than the thrombotic one. The overall length of the manuscript is quite good. I think one of the most important strength points of this review is the good literary overview, which provide a well insight in most recent findings. I suggest the authors to go through the following article recently published, that may help improving “Optimal Antithrombotic Regimen in Patients with AF Undergoing PCI” paragraph (Eyileten C, Postula M, Jakubik D, et al. Non-Vitamin K Oral Anticoagulants (NOAC) Versus Vitamin K Antagonists (VKA) for Atrial Fibrillation with Elective or Urgent Percutaneous Coronary Intervention: A Meta-Analysis with a Particular Focus on Combination Type. J Clin Med. 2020;9(4):1120. Published 2020 Apr 14. doi:10.3390/jcm9041120). Minor spell check is required.

Reviewer 4 Report

In the review article entitled „Antithrombotic Strategy for Patients with Acute Coronary Syndrome: A Perspective from East Asia” by Dr Numasawa et al., the Authors provide a comprehensive literature review of a problem of dual anti-platelet therapy in acute coronary syndrome, focusing on patients from East Asia region or Asian origin. In my opinion it is a very valuable work since undoubtedly, as it was demonstrated by the Authors, there are significant differences between ethnic groups in terms of many factors affecting efficiency and safety of such the therapy. It is also true that many studies seem to not to take into account ethnic differences among patients. The article is well written, I have only a couple of minor comments listed below.

Minor comments

Line 170: Antithrmbotic -> Antithrombotic

Line 240: I think that in a case of a lack of statistical significance, giving detailed p value (p=0.499) should be omitted. The same for line 274, 277, 278.

Lines 260-261 It is difficult to assess whether this incidence were significantly lower since no p values are shown.
